# Enhancing Burn Recovery: A Systematic Review on the Benefits of Electrical Stimulation in Accelerating Healing

**DOI:** 10.3390/ebj6020021

**Published:** 2025-05-05

**Authors:** Dale O. Edwick, Kerry L. Burns, Lara N. Buonvecchi, Xiaolu Wang, Audrey M. Lim, Dale W. Edgar

**Affiliations:** 1School of Allied Health, Faculty of Health Science, Curtin University, Bentley, WA 6102, Australia; 2State Adult Burn Unit, Level 4, Fiona Stanley Hospital, Murdoch, WA 6150, Australia; 3Physiotherapy Department, Fiona Stanley Hospital, Murdoch, WA 6150, Australia; 4School of Health Sciences and Physiotherapy, Faculty of Medicine, Nursing, Midwifery and Health Sciences, The University of Notre Dame Australia, Fremantle, WA 6160, Australia; 5Fiona Wood Foundation, Fiona Stanley Hospital, Murdoch, WA 6150, Australia; 6Institute for Health Research, The University of Notre Dame Australia, Fremantle, WA 6160, Australia; 7Burn Injury Research Unit, University of Western Australia, Crawley, WA 6009, Australia

**Keywords:** acute burn, wound healing, adult, routine wound care, healing time, re-epithelization

## Abstract

Prolonged healing time of acute burn wounds is associated with increased pain, infection, risk of scarring, poorer mobility and higher financial and emotional burden. Electrical stimulation (ES) reduces healing time in chronic wounds; however, its reported use on acute burn wounds is limited. This systematic review (SR) aimed to evaluate the relative benefit of ES compared to routine wound care on the healing time of acute burn wounds in adults. The online databases queried included Cochrane Database of SR’s, MEDLINE, EMBASE, PUBMED and CINAHL. The search criteria included RCTs involving the application of ES of varying voltage, duration and modality in acute burn patients aged ≥18 years. The primary outcome investigated was days to burn wound closure, while the secondary outcomes included edema and infection. Four RCTs were discovered, involving a total of 143 participants with a mean age 35.5 years. Two RCTs demonstrated (a) 36% (2.6 days) reduction in time to wound closure with ES (*p* < 0.001); and (b) significant reduction in wound area with ES (11.2 ± 3.2 cm^2^, *p* < 0.001) compared to controls at 21 days. Two RCTs found ES promoted better wound-healing environments, reducing edema, bacterial infection, and biofilm. This review highlighted low-risk wound-healing benefits with ES as a feasible adjunct to routine burn care.

## 1. Introduction

Expedient management of the acute burn wound with good pain management optimizes clinical outcomes [1,2,3,4,5,6]. Burn wounds are further complicated by edema due to increased vascular permeability and a greater inflammatory response than other wound types [7,8,9]. Prolonged burn-wound-healing time is associated with an increased risk of scarring [10,11]. Scarring associated with burn injury profoundly impacts quality of life due to pain and joint movement restriction and associated poor body image results in deterioration of psychological outcomes [2,12,13,14,15,16]. Improvement in acute-burn-wound-healing time directly influences physical, emotional and psychological recovery as it correlates with functional capacity, mobility, scar flexibility, ability to work, occupational performance, personal independence and mental health [5,17]. Thus, due to the numerous potential benefits, research to optimize burn wound healing and reduce wound infection is warranted and ongoing [18].

The use and facilitation of electrical currents to enhance skin wound healing has been explored since the early 19th century [19]. In the 1800s, Mateucci concluded there was a potential difference between injured and intact skin [20]. Cells are surrounded by a plasma membrane that functions through direct current ion exchange [21]. Further, a potential difference in voltage exists between the stratum corneum and the dermis, where the epidermis is proposed to function as a “skin battery” by generating physiological microcurrents through the active transport of sodium and chloride ions [8,22,23]. Epithelial trauma disrupts this natural transcutaneous potential, generating an endogenous electrical field [8].

The loss of electric potential stimulates cell migration of fibroblasts, keratinocytes, and macrophages to the wound site in a process called galvanotaxis to aid in tissue repair and re-epithelization [22,23,24]. The application of electrical stimulation (ES) has been found to enhance the electrical potential of wounds, thereby aiding the wound-healing process [25,26]. ES creates currents of varying intensity and duration in the form of electrical pulses or microcurrents, which replicate the endogenous electrical fields normally produced during wound healing [27]. The use of electrical stimulation may imitate the endogenous electrical field, enhancing the motility of epithelial cells, thereby accelerating their migration and the healing process [25,28]. Not insignificantly, ES, particularly transcutaneous electrical neural stimulation (TENS), provides acute pain relief and may influence neurological recovery following burn injury [29,30].

ES may also reduce infection, improve cellular maturity and accelerate wound healing [31]. In the absence of intact, healthy skin, the currents generated by ES stimulate angiogenesis and tissue perfusion [32]. During the inflammatory phase, ES increases vasodilation and vascular permeability while inhibiting bacteria proliferation [33,34]. ES has also been demonstrated to shorten the inflammatory phase, promoting migration, proliferation and differentiation of keratinocytes during the proliferative phase while contributing to a more organized alignment of collagen fibrils, with increased conversion of collagen from type III to type I, during the remodelling phase [33,35,36,37].

ES has been shown to increase the rate of healing in chronic wounds by increasing perfusion, controlling bacteria growth and increasing fibroblast migration [31]. ES was shown to be effective in accelerating wound healing in multiple small case studies and clinical trials [35] through the mechanism of reinvigorating ionic flow, which underpins and benefits tissue healing, but which is slowed or stopped in chronic wounds [38,39]. Additionally, electric fields have been demonstrated to affect stem cell differentiation and stromal cell migration, which may facilitate tissue regeneration [40,41]. However, the clinical adoption of ES therapy for acute burn wounds remains limited, largely due to a lack of robust evidence supporting the efficacy of the individual modalities [42]. The endogenous electrical field of burn wounds has been shown to be considerably weaker than other wounds, causing disruptions to wound-healing time [42]. Varying protocols for ES have been reported in chronic wound studies, and the lack of homogeneity in application and methodology means that translation into clinical practice has been slow [43].

The primary aim of this review was to evaluate the effectiveness of ES as an adjunct to routine wound care for accelerating acute burn wound healing when compared to routine wound care alone. The primary outcome investigated was time to re-epithelization, with secondary outcomes including edema management and a reduction in pain and infection.

## 2. Methods

The purpose of this systematic review was to evaluate the effectiveness of ES on acute burn wounds in adults. This review was conducted using the JBI methodology for systematic reviews and reported according to the Preferred Reporting Items for Systematic Reviews and Meta-Analyses (PRISMA) [44,45].

*Inclusion criteria:* Studies meeting the following criteria were included: Population: trials including people aged ≥ 18 years old with a history of acute burn injury (≤10 days) of any depth and size were included; study designs: RCTs, case series studies and case reports studies, all of which were published in English. Only human studies were included. Intervention: studies using ES of any voltage and duration (including ES modalities such as TENS, wireless microcurrent, and electroceutical dressing). Comparison: trials including one or more control groups with the intervention of routine care were included. Outcome: trials including at least one measure of wound healing: time to heal, infection, edema.

*Exclusion criteria:* Animal studies were excluded, as well as studies examining a pediatric population. Studies that performed interventions on non-acute/chronic burns were excluded. Non-interventional studies, as well as those where the intervention involved ultrasound, transcranial direct current stimulation, hydrogels using electrical stimulation, or mixed-mode stimulation techniques, including combinations of light and radio-frequency, such as electro-photobiomodulation, were excluded. Studies that focused on the healing rate of skin graft donor sites, or those that examined participants with electrical burns, were also excluded. Studies without a comparator or control group or that did not measure wound healing as an outcome were omitted. In that, studies that primarily examined pruritus or pain as the outcome were excluded. Studies lacking full text reports, such as conference papers, abstracts only, and other systematic reviews were excluded. Finally, non-English papers were excluded from this review.

### 2.1. Search Strategy

A COCHRANE database search was performed, resulting in no previous studies addressing the objective of this review. Four databases were then systematically searched from their earliest record to capture all related studies: CINAHL from 1982, EMBASE from 1974, MEDLINE from 1966, and PUBMED from 1966. The databases were last searched on 12 August 2024. The search terms were grouped into two concepts: burns and electrical stimulation. The searches were conducted using the terms “burns [MeSH] explosion” OR “title or abstract (burns or burn or burnt or burned)” OR “keyword (burns or burn or burnt or burned)” in all but CINAHL database. Electrical stimulation search terms included “electrical stimulation [MeSH] explosion” OR “electrical stimulation therapy [MeSH] explosion” OR “functional electrical stimulation” OR electrotherapy” OR “transcutaneous electrical nerve stimulation” OR “electric* stimulation or e-stim”. Some terms varied depending on the database used.

Further terms and the search strategy are documented in Appendix A. The results of acute burn searches and electrical stimulation searches were combined using the Boolean operator AND. The full search strategy is documented in Appendix A. Four reviewers (KB, LB, XW, AL) independently screened half the articles each, by abstract and title, as the first stage of this review such that all possible reports included were reviewed by two people. A third reviewer resolved conflicts. Two independent reviewers then reviewed the full text of potentially included articles, and conflicts regarding the exclusion of articles were resolved through a discussion between all four reviewers. The reviewers manually searched the reference lists of the retrieved studies, but did not identify any further articles. Grey literature repositories were searched, and no additional articles were identified.

### 2.2. Assessment of Methodological Quality

The included studies were assessed using the revised JBI critical appraisal tool for the assessment of risk of bias for randomized controlled trials [45]. The reviewers rated the studies using a three-level rating of low risk, high risk, and unclear for 13 questions. The areas assessed included selection and allocation bias, intervention administration bias, reliability of outcome measurement, participant retention bias, and statistical validity. A third reviewer resolved any disagreement. No studies were excluded during this process.

### 2.3. Data Extraction

Data were extracted by two independent reviewers and any disagreements were resolved through discussion with a third reviewer. The extracted data included study characteristics (author, year, country, study type, number of participants, burn size and type, aim, previous treatment, routine care); primary outcome (type of electrical stimulation, intensity, frequency, pulse duration, time of intervention, outcome measure, key results); secondary outcomes (outcome, outcome measure, key results); and adverse effects.

## 3. Results

### 3.1. Study Selection

Electronic searches using key terms yielded 1885 articles, which were exported to Covidence (Covidence systematic review software, Veritas Health Innovation, Melbourne, Australia, 2024), and 612 duplicates were removed (see Figure 1). Four reviewers each screened 1273 articles. In the second stage, a full text screening of the remaining 49 articles was performed by two reviewers independently. Reference lists were manually searched, resulting in no extra inclusions. Four studies met the inclusion criteria and were included in this systematic review. All study designs were randomized control trials (RCT) with varying interventions reported [46,47,48,49].

### 3.2. Study Characteristics

A total of 143 adults participated in the trials, with a mean age of 35.5 years. There was an average of 68% males and 32% females across three trials, with one trial unspecified [48]. Of the 143 participants, 104 had thermal wounds including scald wounds, 5 electrical, 1 chemical, 1 explosion, 1 degloving, and 30 unspecified (see Table 1). Two trials were undertaken in the U.S.A [46,48], one in Australia [47], and one in Egypt [49].

Treatment parameters of voltage, frequency, and duration of ES varied across all trials (see Table 2). All interventions were compared to a control group receiving routine care, and one study compared with a third group receiving negative pressure wound therapy [49].

### 3.3. Risk of Bias in Studies

The methodological quality of the studies is presented in Figure 2. The studies included were of good quality overall. Selection bias was generally low across the studies, but was unclear in one study [49]. There was overall a high risk of performance bias across all but one study due to the nature of the intervention. In all studies, those delivering the intervention were not blind to the treatment due to the mode of applying electrical stimulation; however, due to the nature of the treatment, lack of blinding is unlikely to impact results. Two studies adjusted for performance bias with all patients recruited having incurred burn wounds to separate body parts, where the patient therefore received both the intervention and control treatment [46,47]. Chan et al. [46] utilized intervention and routine care dressings with the same appearance; it is unclear if the participants were blinded to the intervention. Attrition and measurement bias was low across three studies but unclear in one study [49]. Confounding factors were managed in all studies by the exclusion criteria, including exclusions of participants with health conditions. Statistical conclusion validity was strong across all studies.

### 3.4. Effects of Interventions

Wound-healing rate measurement and outcome: Each study utilized a different method for assessing wound healing (see Table 2). Chan et al. [46] used photographic analysis software (Silhouette Connect, Aranz Medical Ltd., Christchurch, NZ) to measure wound closure percentage, transepidermal water loss (TEWL) and scar quality (Vancouver Scar Scale and Patient/Observer Scar Assessment Scale). There were no significant differences between ES and control groups for wound closure and scar outcomes.

Edwick et al. [47] assessed wound-healing using clinical photographic analysis by a burn surgeon blinded to the intervention. The intervention and control wounds both recorded a mean time to re-epithelialize of 19 days (*p* = 0.371) due to the relatively small (1.61 ± 1.29% TBSA) partial thickness wounds. The acute surgical management of the wounds in 20/30 patients in this study aimed to achieve rapid closure regardless of intervention, which may contribute to the similarity in re-epithelization. This study also utilized bioimpedance spectroscopy (BIS) to measure other wound parameters. The BIS parameter of Phase Angle (PhA) was validated as a measure of wound re-epithelialization. While PhA increased significantly in both wounds (*p* < 0.001), the PhA within the stimulated wound increased at a faster rate (stimulation × days post stimulation interaction, X2 (1, N = 907) = 9.49, *p* = 0.002) [47].

Huckfeldt et al. [48] assessed wound healing through digital photography. There was a significant difference in wound closure between the microcurrent group (4.62 ± 1.04 days) and the control group (7.23 ± 1.83 days) (*p* < 0.001). The ES intervention showed a 36% decrease in time to wound closure.

Ibrahim et al. [49] used the metric graph paper method to measure wound surface area. There was no statistically significant difference (*p* ≤ 0.05) in wound surface area reduction between negative pressure wound therapy (NPWT) (4.2 ± 2.7 cm^2^) and microcurrent ES (MES) groups (2.9 ± 2.3 cm^2^) at day 21; however, the MES group showed greater reduction. Wound surface areas were significantly reduced in the MES and NPWT interventions compared to the control group (11.2 ± 3.2 cm^2^, *p* < 0.001). Both MES (27.1 ± 5.9 days) and NPWT (25.1 ± 5.1 days) interventions had significantly lower hospital length of stay than the control group (41.2 ± 8.7 days, *p* < 0.001).

The secondary outcomes of wound infection and edema measures are presented in Table 3. Chan et al. [46] found a 72% reduction in biofilm with ES after seven days, compared with a 48% decrease in the standard care control group (*p* < 0.05). Furthermore, the grafted subgroup showed an 85% reduction in biofilm compared to the control group (46%, *p* < 0.001). Ibrahim et al. [49] similarly demonstrated lower mean bacterial counts at 21 days with MES (2.31 ± 0.89 colonies/mL) and NPWT (2.27 ± 0.74 colonies/mL) than the control group (4.30 ± 0.24 colonies/mL, *p* < 0.001). Edwick et al. [47] demonstrated that the impedance of localized wound edema (R0) increased at a faster rate in the stimulated wound, indicating a faster rate of edema reduction (stimulation × days post stimulation interaction, X2 (1, N = 907) = 5.06, *p* = 0.024).

## 4. Discussion

This review showed that ES reduces healing time and is beneficial to the acute burn wound environment. Prolonged wound healing remains a significant complication of burn injury, and accelerating the time to heal will improve patient outcomes and reduce healthcare costs [2,11]. Wound healing in three of the studies measured time to achieve wound closure based on assessment of digital photography of the wounds by experienced burns surgeons blinded to the intervention [46,47,48]. Digital photography is a valid objective measure of wound healing, with increased reliability noted with experienced assessors, and is used clinically to track and record wound healing as part of the electronic medical record [50,51,52,53]. Huckfeldt et al. [48] observed a significant reduction in time to achieve wound closure, with ES resulting in a 36% decrease in time compared to the control group. Additionally, Ibrahim et al. [49] reported a non-significant reduction in wound surface area with MES. However, there were no significant differences in time to heal in the studies by Chan et al. [46] and Edwick et al. [47]. Data synthesis of the studies in this review was not possible as each study reported different outcomes while investigating different ES modalities. In addition to accelerated wound healing [48,49], the findings of the RCTs included in this review demonstrated a promotion of optimal healing environments through improved rate of edema reduction [47] and reduction in bacterial load [46,49].

Four different ES modalities were investigated in the studies. Huckfeldt et al. [48] placed one electrode on the wound dressing and the other on healthy skin, and Ibrahim et al. [49] did not specify. Edwick et al. [47] placed ES electrodes on either side of the burn wound on healthy skin. Chan et al. [46] used a wireless electroceutical dressing (WED), branded as Procellera by Vomaris Innovations, consisting of a polyester dressing embedded with a geometrically printed matrix of silver and zinc nanoparticles, placed directly on the wound. Furthermore, Edwick et al. [47] used Bio-Flex stimulation electrodes from the ActivMed TENS device by ActiveLife Technologies. An electrode was positioned on intact skin proximal to the wound, with a second electrode placed 30 mm distal to the wound, generating a current across the wound bed. Huckfeldt et al. [48] used a continuous direct anodal microcurrent generated by placing electrically conductive perforated carbon vinyl tape over silver nylon fabric, connected to a microcurrent generator via a standard electrode lead. A return electrode was positioned on intact skin—distal to the graft or donor site. Ibrahim et al. [49] generated microcurrent electrical stimulation (MES) using a modified square biphasic pulsed waveform with a 50% duty cycle, alternating negative and positive polarity every second.

The variation in reported outcomes is likely be due to the differences in ES modalities, which included microcurrent ES, direct anodal microcurrent, and wireless electroceutical dressing. Rabbani et al. [54] report in their review that direct current (DC) promotes fibroblast proliferation and collagen secretion, as well as reducing infection rates, while alternating current (AC) has been associated with enhanced angiogenesis and collagen expression in wound healing. Direct current has been found to be effective in reducing bacterial load by disrupting the bacterial membrane and blocking the proliferation of bacterial cells, and indirectly through pH changes, which render the wound inhospitable to bacteria [33]. Therefore, AC may be most beneficial if applied during the acute remodelling phase. A combination of both current types may produce optimal healing effects [54].

This review also highlighted the lack of consensus in wound-healing measurement techniques in the literature. The outcome measures in the included studies highlighted other benefits of ES in the acute burn wound. Chan et al. [46] also assessed wound healing using TEWL, a measure of the health of the skin barrier, which is a reliable biomarker of wound healing [55]. Edwick et al. [47] obtained serial measures of the wounds using BIS, and the BIS parameter Phase Angle (PhA) was validated against the wound photography as a measure of wound healing in acute burn wounds, which has been demonstrated previously in chronic wounds [52]. PhA is the arctangent of the recorded measures of reactance and resistance, which is expressed in degrees. As the bioimpedance frequency increases, the current flow overcomes the natural capacitive impedance of the cell wall, allowing the current to pass through the cells. This creates a delay in the flow of current with respect to voltage, known as PhA. PhA is believed to be a measure of the health of the cell, as healthier cells have thicker walls and are heavier, resulting in an increased time for the current to traverse the cell, resulting in an increased PhA [54,55]. Quantitative methods of wound assessment minimize the subjectivity and variability depending on the experience of the assessor, and BIS is a promising modality for assessing wound status [51].

This systematic review reflects the applicability of reported findings of studies investigating ES in healing of other wound types. Zhao et al. [53] reported improved healing times in chronic wounds with the application of ES. Similarly, Ofstead et al. [56] examined continuous electrical microcurrent on acute and hard-to-heal wounds, confirming that microcurrent ES significantly decreased healing times in both wound types when compared to standard care. Luo et al. [26] and Rajendran et al. [31] report reduced wound size and bacterial growth, and increased wound blood perfusion, fibroblast migration, angiogenesis, and keratinocyte activity with the application of ES in both acute and chronic wounds. Wireless microcurrent ES also improved blood flow, haemoglobin and oxygen levels in partial thickness burn wounds [57]. While this review only investigated adult patients with acute burn wounds, a recent study involving pediatric patients evaluating the effectiveness of wireless microcurrent stimulation (WMCS) as an adjunct to standard wound care demonstrated shorter median healing time when compared to the control group, despite the wound size being two times larger (2.78% vs. 1.39% TBSA) in the intervention group [58].

## 5. Conclusions

This review included four RCTs involving 143 patients, indicating that the current evidence base for ES in acute burn care is limited. Despite this, this review illustrates that the potential low-risk intervention provides benefits of ES in acute burn injury due to concurring results from multiple study sites. Thus, this systematic review highlights the potential of ES to adjunct to routine care during burn treatment, improving wound healing rate, and reducing pain, wound edema and bacterial load. More robust evidence is necessary to establish the optimal parameters of ES therapy in clinical practice through standardized methods, measurements and outcome protocols and larger clinical trials. These findings are still relevant to the advancement of ES research in its application to improve patient outcomes and guide future clinical practice in burn management.

## Figures and Tables

**Figure 1 ebj-06-00021-f001:**
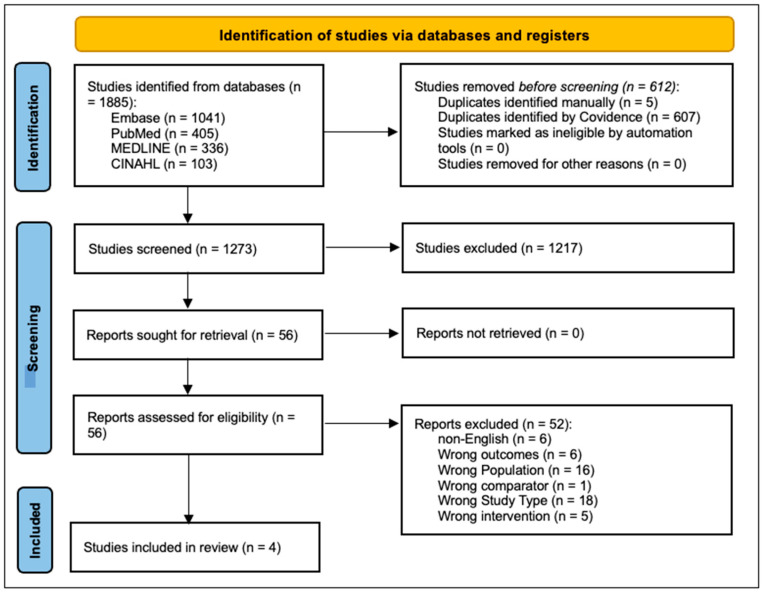
PRISMA 2020 flow diagram for new systematic reviews which included searches of databases and registers only.

**Figure 2 ebj-06-00021-f002:**
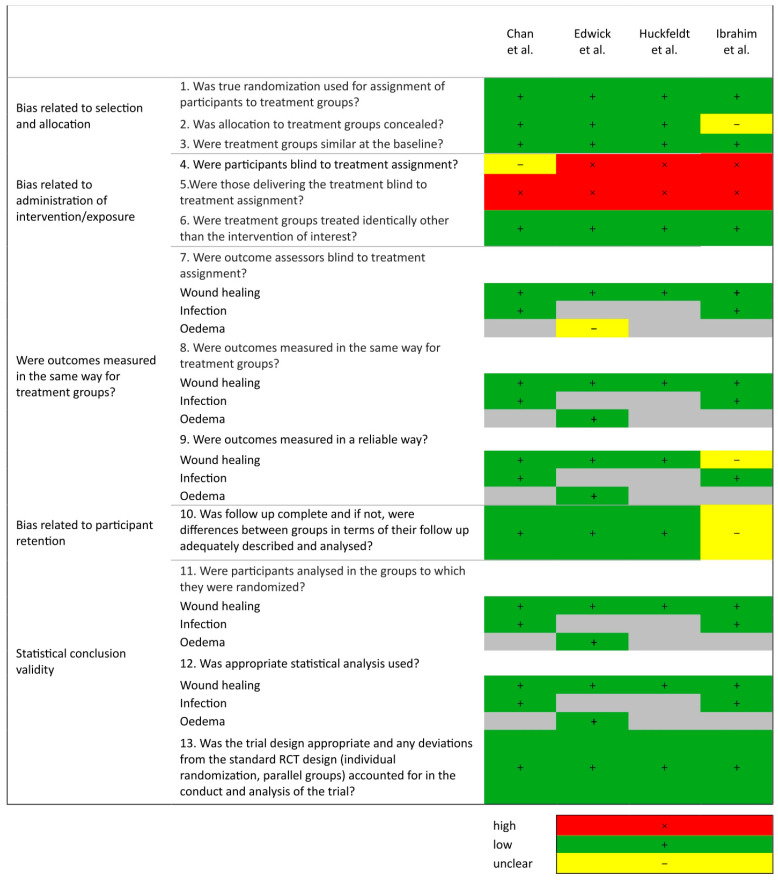
Revised JBI critical appraisal tool for the assessment of risk of bias for randomized controlled trials, [46,47,48,49].

**Table 1 ebj-06-00021-t001:** Study characteristics.

Author/Year Country	Study Design	N	Age (Mean + Range)	M/F Ratio (%)	Baseline Burn Size	Burn Type	Aim	Treatment Previous to Intervention	Routine Care
Chan et al. (2024) U.S.A. [42]	RCT	38	Mean 45 23–84	76.32/23.68	1 × Burns wounds ≥ 300 cm^2^, or 2 × Burns wounds ≥ 150 cm^2^	29 thermal 5 electrical 1 chemical 1 explosion/blast 1 degloving	To evaluate the effect of wireless electroceutical dressing (WED) compared with routine dressings to reduce biofilm on burn wounds.	Treatments deemed necessary which included debridement, negative pressure wound therapy (NPWT), and skin grafting.	Standard care included but was not limited to silver nylon, SSD ointment, bacitracin, xeroform, 5% sulfamylon solution, and Manuka honey.
Edwick et al. (2022) Australia [43]	RCT	30	Median 32.5 18–72	80/20	0.1–5.5% TBSA	5 scalds 5 contact 14 flash/flame 2 friction 3 chemical 1 radiation	To measure wound healing in acute minor burn injuries using Bioimpedance Spectroscopy (BIS), where electrical stimulation was used as a treatment to reduce the presence of edema.	Surgical intervention was required in 20 of the patients, including dermabrasion and ReCell (n =11), and a combination of split-thickness skin graft ±ReCell (n =9). All wounds were debrided and equivalent surgery technique was applied to each wound of the paired study wounds.	Patients received proactive oedema treatments from their therapists, which including exercise and compression therapies, and education and assistance with positioning for elevation of affected limbs, in addition to routine dressing changes.
Huckfeldt et al. (2007) U.S.A. [44]	RCT	30	Mean 40 18–68	Not Specified	Not Specified	30 thermal	To test the effect of continuous direct anodal microcurrent on wound closure time when applied to silver nylon wound contact dressings after split-thickness skin grafting.	Split-thickness skin grafting.	The control group were dressed at the graft site using silver nylon fabric moistened with sterile water in direct contact with the wound surface and covered with gauze and an elastic bandage.
Ibrahim et al. (2019) Egypt [45]	RCT	45	Mean 26.63 20–40	53.33/46.67	31.26% TBSA	25 flame 20 scalds	To compare the efficacy of negative pressure wound therapy (NPWT) with that of microcurrent electrical stimulation (MES) on wound surface area, length of stay (LoS) and colony count of wounds in patients with burns.	N/A	The control group (as well as the NPWT and MES groups) received standard medical wound care (wound dressing, nursing care and pain relief medication) and a routine rehabilitation programme (range of motion exercises, ambulation training, and positioning and stretching exercises) throughout the study period.

**Table 2 ebj-06-00021-t002:** Treatment parameters and wound-healing assessment.

Author	Type of Electric Stimulation	Intensity Current (µA) OR Voltage (V)	Frequency (Hz)	Pulse Duration (μs)	Duration (Per Session)	Time (Days)	Outcome Measure	Key Results
Chan et al. (2024) [42]	Wireless Electrical Dressing (WED) Polyester dressing with a matrix of elemental silver and zinc nano particles that generate a weak electrical field on contact with a conductive medium (e.g., hydrogel)	~1 V	N/A	N/A	24 h/day	7 days	Wounds were photographed and assessed with SilhouetteStar camera and software to measure wound closure %. Transepidermal water loss (TEWL) (g/m2/h) using DermaLab TEWL Probe Patient/Observer Scar Assessment Scale (POSAS) and Vancouver Scar Scale (VSS)	WED did not significantly impact the long-term outcome of wound healing.
Edwick et al. (2022) [43]	Electrical stimulation delivered via Bio-Flex stimulation electrodes (ActivMed stimulation device) on intact skin—one inferior and one superior to wound	12–30 V	6–12 Hz	200 μs	>20 h/day (recommended by device manufacturer)	10–14 days	BIS measurements of control and intervention wounds. Photographic analysis (serial wound photographs). Burn Surgeon analysed photographic evidence to determine percentage of re-epithelialization (time to heal)	No significant difference (*p* = 0.371) in time to heal (days) between control group and stimulation group.
Huckfeldt et al. (2007) [44]	Moistened silver nylon fabric covered with gauze with addition of continuous direct anodal microcurrent application	5.0 V 50–100 μA	direct anodic microcurrent	constant	24 h/day	until 95% wound closure	Time to 95% wound closure was measured using digital photography.	The study group experienced a 36% reduction in time to wound closure (mean of 4.6 days) as compared to the control group (mean of 7.2 days), (*p* < 0.05)
Ibrahim et al. (2019) [45]	Microcurrent Electrical Stimulation (MES) delivered through a modified square biphasic pulsed waveform	300 μA	10 Hz	constant	1 h/day	3 days a week for 3 weeks (21 days), or until wound closure	Wound surface area was measured using the metric graph paper method The length of stay (LoS) was also recorded; the number of days that the patient stayed in the hospital until wound closure.	MES was found to be effective in decreasing wound size within the first 10 days after injury. On day 21, MES showed the greatest reduction in wound surface area, compared to NPWT and the control group (*p* =< 0.001). There was no significant difference in LoS between the MES and NPWT groups, but the statistically significantly highest mean LoS was shown in the control group (*p* < 0.001)

**Table 3 ebj-06-00021-t003:** Secondary outcomes.

Author	Outcome	Outcome Measure	Key Results
Chan et al. (2024) [42]	Biofilm and bacterial infection	Scanning electron microscopy (SEM) quantification imaging. SEM biofilm grading used a 0–3 scale.	WED significantly decreased biofilm by 72% compared with 48% decrease from SoC treatment. WED decreased infection by 62% compared with 52% from standard of care treatment. Non-grafted burn wounds had significantly lower levels of biofilm detected in WED treated wounds than SoC.
Edwick et al. (2022) [43]	Evaluate BIS raw variables as a valid measure for edema and wound responses.	Repeated serial BIS was used to measure wound healing using the following 4 measures. *Ro* = Localized wound edema impedance. *Ri* = Localised intracellular fluid impedance. *R∞* = Total fluid within the wound impedance. PhA = Phase angle (°) which is the arctangent measurement of resistance and reactance.	BIS can be interpreted as a direct physiological measure of cellular architecture and function (tissue and cellular health) *Ro* = increased at a faster rate in the stimulated wound when compared to the control wound (*p* = 0.024) with faster reduction of ion rich edema in the intervention wound. *Ri* = increased significantly (*p* = 0.001) but no difference in change between wounds. *R∞* = increased significantly in both wounds (*p* < 0.001), with increased rate of change in the intervention wound (*p* = 0.045). PhA = increased significantly in both wounds (*p* < 0.001), with increased rate of change in the intervention wound (*p* = 0.002).
Huckfeldt et al. (2007) [44]	N/A	N/A	N/A
Ibrahim et al. (2019) [45]	Infection	Bacterial Count (Colonies)	At day 21, both the NPWT and MES groups had a statistically significant lower mean bacterial count while the control group revealed a statistically significant higher mean bacterial count.

## Data Availability

All data are available within this manuscript.

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
