# Peer review of "Enhancing Burn Recovery: A Systematic Review on the Benefits of Electrical Stimulation in Accelerating Healing"

_2673-1991, 2025, doi:10.3390/ebj6020021_

Round 1
Reviewer 1 Report
Comments and Suggestions for Authors
This manuscript examines the application of electrical stimulation (ES) as a supplementary treatment for acute burns, addressing a notable gap in systematic evidence within this domain and presenting an innovative approach.
- Introduction
1)The introduction lacks a comprehensive discussion on the "differences in electrophysiological mechanisms between acute burns and chronic wounds,". It is advisable to incorporate relevant literature to substantiate this point.
2)Expand the conversation about the state of acute burn care today (such as the shortcomings of current treatments) and emphasize the necessity of ES.
- Discussion and conclusions
1)It explains why certain investigations have revealed notable effects when combined with the molecular processes by which ES stimulates cell migration (e.g., fibroblasts, keratinocytes) (e.g., calcium signaling pathways, growth factor release).
2) Mechanistic explanation: A more thorough examination of how ES reduces bacterial load is warranted. Furthermore, the distinct impacts of various current types (DC versus AC) on the healing phases.
It is advisable to accept it following the incorporation of the aforementioned details.
Author Response
Comments 1: The introduction lacks a comprehensive discussion on the "differences in electrophysiological mechanisms between acute burns and chronic wounds,". It is advisable to incorporate relevant literature to substantiate this point.
Response 1: Thank you for this comment. Statements added and references #38-41 to support. These changes can be found on Page 2 - Lines 80-83. All changes are in red text (Please see attachment).
Comments 2: Expand the conversation about the state of acute burn care today (such as the shortcomings of current treatments) and emphasize the necessity of ES.
Response 2: Statement added, supported by additional reference #18. These can be found on Page 2 - Lines 46-48.
Comments 3: It explains why certain investigations have revealed notable effects when combined with the molecular processes by which ES stimulates cell migration (e.g., fibroblasts, keratinocytes) (e.g., calcium signaling pathways, growth factor release).
Response 3: thank you for the accurate summary.
Comments 4: Mechanistic explanation: A more thorough examination of how ES reduces bacterial load is warranted. Furthermore, the distinct impacts of various current types (DC versus AC) on the healing phases.
Response 4: Thank you, statements added to expand on findings in reference #33. These can be found on Page 10 - lines 287-290.
Reviewer 2 Report
Comments and Suggestions for Authors
good study, but why did he do it for adults only, why not for pediatrics?
Author Response
Comments 1: good study, but why did he do it for adults only, why not for pediatrics?
Response 1: Thank you for the positive review comment. In relation to the challenges of application of ES in children, the scope of this review was kept to adults. That said, the reviewer makes a good pitch for a future review.
Reviewer 3 Report
Comments and Suggestions for Authors
Overall: Interesting article, well written and structured. Small corrections have to be done.
- Title: It's confusing, I suggest something more appealing like: "Enhancing Burn Recovery: A Systematic Review on the Benefits of Electrical Stimulation in Accelerating Healing"
- English grammar, sentence structure, and article design: Well written and structured article. All the ideas are put across in a clear and interconnected way.
- Abstract: The abstract is good and has everything that is needed. Everything is connected and well-written. However, the keywords can't be the same as the ones in the title, please review that. Furthermore, the graphic that comes after, I like to point out that it´s an amazing graphic, however I´m confused: Is this the abstract graphic or a figure for the text? Because if it is the second, this should have a legend and be cited on the text, and only after the introduction
- Introduction: Again, the introduction is good and has everything that is needed. Nothing to point out.
- Methods: Once again, the Methods are good and have everything that is needed. However, I suggest an alteration, in line 130-131: “Further terms and search strategy are documented in Appendix A. The results of acute burn searches 130 and electrical stimulation searches were combined using the Boolean operator AND.”
- Results: This section is good and has everything that is needed. But I have to point out the unreadable tables 1,2, and 3 – I suggest simplifying and shortening the text in the table and putting it bigger.
- Discussion: Again, this part is good and has everything that is needed. However, I suggest a rewriting of this part in a way to use the connected terms throughout the text, like: Furthermore, Additionally, On the contrary, etc - only to connect more the text.
- Conclusion: Again, this section is good and has everything that is needed. I have nothing to point out.
- References: I have nothing to point out.
Author Response
Comments 1: Overall: Interesting article, well written and structured. Small corrections have to be done.
Response 1: Thank you to Reviewer 3 for the positive comment
Comments 2: Title: It's confusing, I suggest something more appealing like: "Enhancing Burn Recovery: A Systematic Review on the Benefits of Electrical Stimulation in Accelerating Healing"
Response 2: Thank you for the suggestion. Title adjusted. Page 1 Line 1-3
Comments 3: English grammar, sentence structure, and article design: Well written and structured article. All the ideas are put across in a clear and interconnected way.
Response 3: Thank you for the positive comment.
Comments 4: Abstract: The abstract is good and has everything that is needed. Everything is connected and well-written. However, the keywords can't be the same as the ones in the title, please review that.
Response 4: Thank you for the suggestion, new keywords added. Page 1 Line 33-34
Comments 5: Furthermore, the graphic that comes after, I like to point out that it´s an amazing graphic, however I´m confused: Is this the abstract graphic or a figure for the text? Because if it is the second, this should have a legend and be cited on the text, and only after the introduction
Response 5: The infographic is not a figure in text. As per journal requirements, it is placed ahead of the Abstract. Apologies for the confusion.
Comments 6: Introduction: Again, the introduction is good and has everything that is needed. Nothing to point out.
Response 6: Thank you for the positive comments.
Comments 7: Methods: Once again, the Methods are good and have everything that is needed. However, I suggest an alteration, in line 130-131: “Further terms and search strategy are documented in Appendix A. The results of acute burn searches 130 and electrical stimulation searches were combined using the Boolean operator AND.”
Response 7: Thank you for the suggestions, adjustments made as requested. Page 3 Line 133-135
Comments 8: Results: This section is good and has everything that is needed. But I have to point out the unreadable tables 1,2, and 3 – I suggest simplifying and shortening the text in the table and putting it bigger.
Response 8: Thank you. We will work with the publisher to improve the visibility of the tables as they are submitted as figures to adhere with formatting requirements.
Comments 9: Discussion: Again, this part is good and has everything that is needed. However, I suggest a rewriting of this part in a way to use the connected terms throughout the text, like: Furthermore, Additionally, On the contrary, etc - only to connect more the text.
Response 9: Thank you for the suggestions, adjustments made as requested. Tracked in red through discussion section.
Comments 10: Conclusion: Again, this section is good and has everything that is needed. I have nothing to point out.
References: I have nothing to point out.
Response 10: Thank you for the positive comments.
Reviewer 4 Report
Comments and Suggestions for Authors
Overall a very well presented review. It clearly follows the appropriate reporting guidelines. The topic is of clinical interest and it was an enjoyable read.
Suggestions:
The abstract could be a little clearer about the primary outcome of the study - this is made very clear in the introduction (see below) so consider adding specific outcomes measured.
One small typographical error shown in red (This needs to be replaced with The)
The primary aim of this review was to evaluate the effectiveness of ES as an adjunct to routine wound care for accelerating acute burn wound healing, when compared to routine wound care alone. The primary outcome investigated was time to re-epithelization, with secondary outcomes including edema management and reduction of pain and infection.
The Figure on Page 2 is somewhat unconventional - it appears to be a poster style Figure to summarise the results. This may be better at the end of the manuscript (depending on the Editors views).
Author Response
Comments 1: Overall a very well presented review. It clearly follows the appropriate reporting guidelines. The topic is of clinical interest and it was an enjoyable read.
Response 1: Thank you to Reviewer 4 for the positive comments.
Comments 2: Suggestions:
The abstract could be a little clearer about the primary outcome of the study - this is made very clear in the introduction (see below) so consider adding specific outcomes measured.
Response 2: Thank you. Wording tweak added to Abstract. please see Page 1 Line 25.
Comments 3: One small typographical error shown in red (This needs to be replaced with The)
Response 3: Thank you, adjusted. Please see Page 2 Line 90.
Comments 4: The primary aim of this review was to evaluate the effectiveness of ES as an adjunct to routine wound care for accelerating acute burn wound healing, when compared to routine wound care alone. The primary outcome investigated was time to re-epithelization, with secondary outcomes including edema management and reduction of pain and infection.
Response 4: Thank you for highlighting. Applied in Abstract.
Comments 5: The Figure on Page 2 is somewhat unconventional - it appears to be a poster style Figure to summarise the results. This may be better at the end of the manuscript (depending on the Editors views).
Response 5: Thank you. The infographic is not a figure in text. As per journal requirements, it is placed ahead of the Abstract. Apologies for the confusion.
Reviewer 5 Report
Comments and Suggestions for Authors
Dear authors,
Thank you very much for considering EBJ for publication of your revised »Electrical stimulation is beneficial as an adjunct to routine care, reducing healing time of acute burns in adult patients: A systematic review«.
Overall, this systematic review provides valuable insights into the potential benefits of ES in acute burn wound healing. Addressing the methodological and interpretive concerns outlined would improve the clarity, reliability and applicability of the findings. You have also nicely demonstrated the purpose of the study and its findings with interesting diagrams.
You might also consider discussing the clinical significance of the digital photography, transepidermal water loss, and bioimpedance spectroscopy metrics (e.g., their reliability, validity, and correlation with functional outcomes), as this might strengthen the practical applicability of your findings.
We have no further comments.
Author Response
Comments 1: Dear authors,
Thank you very much for considering EBJ for publication of your revised »Electrical stimulation is beneficial as an adjunct to routine care, reducing healing time of acute burns in adult patients: A systematic review«.
Overall, this systematic review provides valuable insights into the potential benefits of ES in acute burn wound healing. Addressing the methodological and interpretive concerns outlined would improve the clarity, reliability and applicability of the findings. You have also nicely demonstrated the purpose of the study and its findings with interesting diagrams.
Response 1: Thank you to Reviewer 5 for these positive comments.
Comments 2: You might also consider discussing the clinical significance of the digital photography, transepidermal water loss, and bioimpedance spectroscopy metrics (e.g., their reliability, validity, and correlation with functional outcomes), as this might strengthen the practical applicability of your findings.
Response 2: Thank you for the suggestions, Discussion adjusted - Page 10 line 253-256 and Line 296.
We have no further comments.
Reviewer 6 Report
Comments and Suggestions for Authors
I have carefully reviewed the manuscript. The introduction reflects the research direction, the methodology is properly and reliably conducted, and the results are consistent and clear. However, in the recommendations, it is suggested that reviews should be registered in the PROSPERO database, which can also be done after the study is completed. Despite my positive review, the final decision on accepting the paper is left to the editor-in-chief. I have no further comments on the work. Congratulations and best regards.

Author Response
Comments 1: I have carefully reviewed the manuscript. The introduction reflects the research direction, the methodology is properly and reliably conducted, and the results are consistent and clear. However, in the recommendations, it is suggested that reviews should be registered in the PROSPERO database, which can also be done after the study is completed. Despite my positive review, the final decision on accepting the paper is left to the editor-in-chief. I have no further comments on the work. Congratulations and best regards.
Response 1: Thank you to Reviewer 6 for their positive comments. Unfortunately, PROSPERO will not accept submissions of review protocols in retrospect. We will certainly submit our protocol a priori in future. Thank you for the comment.
Round 2
Reviewer 2 Report
Comments and Suggestions for Authors
this study does not include Electro-photobiomodulation references, which have used electricity with excellent results
Author Response
Comments 1: this study does not include Electro-photobiomodulation references, which have used electricity with excellent results
Response 1: Thank you for this comment. Statement "...or mixed-mode stimulation techniques, including combinations of light and radio-frequency, such as electro-photobiomodulation," added to exclusion criteria - Page 3, line 114-115.
Reviewer 3 Report
Comments and Suggestions for Authors
I must congratulate the authors on thoroughly following all the suggestions in this new version.
Author Response
Comments 1: I must congratulate the authors on thoroughly following all the suggestions in this new version.
Response 1: Thank you for the positive comment.
Reviewer 4 Report
Comments and Suggestions for Authors
Thank you for addressing the responses to the comments on the previous version. This now reads well.
Author Response
Comments 1: Thank you for addressing the responses to the comments on the previous version. This now reads well.
Response 1: Thank you for the positive comment.
Reviewer 5 Report
Comments and Suggestions for Authors
Dear authors,
Thank you very much for considering EBJ for publication of your revised »Enhancing burn recovery: A systematic review on the benefits of electrical stimulation in accelerating healing.«
You have considered our suggestions and made appropriate changes.
We have no further comments.
Author Response
Comments 1:
Dear authors,
Thank you very much for considering EBJ for publication of your revised »Enhancing burn recovery: A systematic review on the benefits of electrical stimulation in accelerating healing.«
You have considered our suggestions and made appropriate changes.
We have no further comments.
Response 1: Thank you for addressing the responses to the comments on the previous version. This now reads well.
Reviewer 6 Report
Comments and Suggestions for Authors
Thank you for your reply
Author Response
Comments 1: Thank you for your reply
Response 1: Thank you for the positive comment.